# Head-to-Head Comparison of Rapid and Automated Antigen Detection Tests for the Diagnosis of SARS-CoV-2 Infection

**DOI:** 10.3390/jcm10020265

**Published:** 2021-01-13

**Authors:** Julien Favresse, Constant Gillot, Maxime Oliveira, Julie Cadrobbi, Marc Elsen, Christine Eucher, Kim Laffineur, Catherine Rosseels, Sandrine Van Eeckhoudt, Jean-Baptiste Nicolas, Laure Morimont, Jean-Michel Dogné, Jonathan Douxfils

**Affiliations:** 1Department of Laboratory Medicine, Clinique St-Luc Bouge, 5004 Namur, Belgium; julie.cadrobbi@slbo.be (J.C.); marc.elsen@slbo.be (M.E.); christine.eucher@slbo.be (C.E.); kim.laffineur@slbo.be (K.L.); catherine.rosseels@slbo.be (C.R.); 2Department of Pharmacy, Namur Research Institute for Life Sciences, University of Namur, 5000 Namur, Belgium; constant.gillot@unamur.be (C.G.); maxime.oliveira@qualiblood.eu (M.O.); laure.morimont@unamur.be (L.M.); jean-michel.dogne@unamur.be (J.-M.D.); jonathan.douxfils@unamur.be (J.D.); 3Department of Internal Medicine, Clinique St-Luc Bouge, 5004 Namur, Belgium; sandrine.vaneeckhoudt@slbo.be (S.V.E.); jeanbaptiste.nicolas@slbo.be (J.-B.N.); 4Qualiblood S.A., 5000 Namur, Belgium

**Keywords:** COVID-19, SARS-CoV-2, antigen testing, manual assay, automated assay

## Abstract

(1) Background: The detection of SARS-CoV-2 RNA in nasopharyngeal samples through real-time reverse transcription-polymerase chain reaction (RT-PCR) is considered the standard gold method for the diagnosis of SARS-CoV-2 infection. Antigen detection (AD) tests are more rapid, less laborious, and less expensive alternatives but still require clinical validation. (2) Methods: This study compared the clinical performance of five AD tests, including four rapid AD (RAD) tests (biotical, Panbio, Healgen, and Roche) and one automated AD test (VITROS). For that purpose, 118 (62.8%) symptomatic patients and 70 (37.2%) asymptomatic subjects were tested, and results were compared to RT-PCR. (3) Results: The performance of the RAD tests was modest and allowed us to identify RT-PCR positive patients with higher viral loads. For Ct values ≤25, the sensitivity ranged from 93.1% (95% CI: 83.3–98.1%) to 96.6% (95% CI: 88.1–99.6%), meaning that some samples with high viral loads were missed. Considering the Ct value proposed by the CDC for contagiousness (i.e., Ct values ≤33) sensitivities ranged from 76.2% (95% CI: 65.4–85.1%) to 88.8% (95% CI: 79.7–94.7%) while the specificity ranged from 96.3% (95% CI: 90.8–99.0%) to 99.1% (95% CI: 95.0–100%). The VITROS automated assay showed a 100% (95% CI: 95.5–100%) sensitivity for Ct values ≤33, and had a specificity of 100% (95% CI: 96.6–100%); (4) Conclusions: Compared to RAD tests, the VITROS assay fully aligned with RT-PCR for Ct values up to 33, which might allow a faster, easier and cheaper identification of SARS-CoV-2 contagious patients.

## 1. Introduction

Currently, the revelation of SARS-CoV-2 RNA through real-time reverse transcription-polymerase chain reaction (RT-PCR) from nasopharyngeal swab samples is considered the standard gold method for the diagnosis of SARS-CoV-2 infection. The targeted genes may include a combination of *N*, *E*, *RdRp*, *orf1a*, and *orf1b* genes [1,2]. However, even if the RT-PCR method is highly sensitive, positive SARS-CoV-2 samples do not always allow definitive conclusions about whether the subject is contagious or not [3]. This can be partly assessed by the cycle threshold (Ct) value, which has been correlated to the amount of viral RNA in the sample, informing on the viral load and infectivity [4].

In addition, in most laboratories, the analysis throughput of RT-PCR testing is limited because it requires a high workload, skillful operators, expensive instrumentation, and crucial biosafety measures [5,6]. In contrast, antigen detection (AD) tests are more rapid, less laborious, less expensive, and require only a comparatively short training period. Currently, there are approximately 100 different AD tests on the market [7]. Head-to-head comparisons of AD test performance for SARS-CoV-2 detection are scarce. The majority of the studies only included few patients, precluding a wide characterization of the “real-life” performance of these devices [3,8,9,10,11,12,13,14,15,16,17,18,19,20,21,22,23]. Recently, a Cochrane study found an average sensitivity of only 56.2% (95% confidence interval (CI): 29.5–79.8%) for rapid AD (RAD) tests [11]. Independent reporting of automated AD test performance remains limited [18].

This study aimed to compare the clinical performance of 4 RAD tests ((i) the Biotical SARS-CoV-2 Ag card, (ii) the Panbio™ COVID-19 Ag Rapid Test Device, (iii) the Coronavirus Ag Rapid Test Cassette, and (iv) the Roche SARS-CoV-2 Rapid Antigen Test)) and one automated AD test (VITROS Immunodiagnostic Products SARS-CoV-2 Antigen test), to the results of the RT-PCR.

## 2. Material and Methods

### 2.1. Patients and Sample Collection

This study was conducted from 17 November to 25 November 2020 at the clinical biology laboratory of the Clinique Saint-Luc (Bouge, Namur, Belgium). Nasopharyngeal samples from 188 patients (104 females (median age = 54 years; min-max: 5–97 years) and 84 males (median age: 57 years; min-max: 1–94 years)) who presented at our institution for SARS-CoV-2 testing were included. One hundred and eighteen (62.8%) were symptomatic patients, and 70 (37.2%) were asymptomatic subjects. In symptomatic patients, the median time since symptom onset was three days (interquartile range (IQR): 2–4 days). Information on the days since the onset of symptoms was collected from medical records and was available for all patients. Nasopharyngeal samples were collected using eSwab liquid preservation medium (Copan Italia, Brescia, Italy) or Vacuette Virus Stabilization (Greiner Bio-One, Kremsmünster, Austria) tubes. The same tube was used for both RT-PCR and antigenic assessments. The study protocol was in accordance with the Declaration of Helsinki and was approved by the Medical Ethical Committee of Saint-Luc Bouge (Bouge, Belgium, approval number B0392020000005).

### 2.2. Analytical Procedures

The 4 RAD tests included were: (i) the Biotical SARS-CoV-2 Ag card (Biotical Health, Madrid, Spain), (ii) the Panbio™ COVID-19 Ag Rapid Test Device (Abbott, Chicago, IL, USA), (iii) the Coronavirus Ag Rapid Test Cassette (Healgen Scientific, Houston, TX, USA) and (iv) the Roche SARS-CoV-2 Rapid Antigen Test (Roche Diagnostics, Basel, Switzerland). Samples identified as positive at both the control line and test line were regarded as SARS-CoV-2 antigen-positive, and samples having only the control line were regarded as SARS-CoV-2 antigen-negative. The automated AD test was the VITROS Immunodiagnostic Products SARS-CoV-2 Antigen test (Ortho Clinical Diagnostics, Raritan, NJ, USA) performed on the VITROS 3600 Immunodiagnostic System (Ortho Clinical Diagnostics). A signal ≥1, as rendered by the analyzer, was considered positive. The target of the 5 AD tests was the nucleocapsid, i.e., the main antigen used in available AD tests on the market [7]. Table 1 summarizes the characteristics and procedures of these AD tests. The viral transport media was used instead of the swab provided in the kits in order to allow the assessment of 5 different AD tests on the same sample. The analyses were performed according to the recommendations of the manufacturers.

Two independent operators (CG and MD) interpreted the results of the four RAD tests. In case of disagreement (i.e., in case of low band intensity), a third-blinded operator (JD) interpreted the result to find a consensus result. All operators were blinded to RT-PCR results and clinical data while performing or interpreting AD tests. The results of the four RAD tests were read after the reaction times recommended by the manufacturers; to note: if the manufacturer recommended reading the result between a certain interval of times (e.g., between 15 and 20 min), two readings were performed at the lowest and highest recommended times.

The RT-PCR for SARS-CoV-2 determination of nasopharyngeal swab samples was performed on a LightCycler^®^ (Roche Diagnostics^®^, Basel, Switzerland)) 480 Instrument II (Roche Diagnostics^®^) using the LightMix^®^ (Roche Diagnostics^®^) Modular SARS-CoV *E*-gene set. Cycle threshold values obtained by RT-PCR were used as a proxy for the viral load. All tests were performed within a maximum of 24 h after specimen collection.

### 2.3. Statistical Analyses

Descriptive statistics were used to analyze the data. Sensitivity was defined as the proportion of SARS-CoV-2 positive samples by the AD test initially categorized as positive by RT-PCR. Specificity was defined as the proportion of samples identified as negative by the AD test initially categorized as negative by RT-PCR. The sensitivity and specificity analyses were also performed by considering the Ct value determined as contagiousness instead of considering all Ct values as true positive results. Contagiousness was defined as the Ct values at which a viral culture has been reported negative, according to the literature (i.e., ≤25, ≤33, and ≤35) [4,8,24,25,26,27,28]. Positive predictive values (PPV), negative predictive values (NPV), and test accuracy at different disease prevalence, i.e., 2, 5, 10, 15, 20, and 25%, were also calculated. A Mann–Whitney test was used to assess differences between groups. A simple linear regression was computed to assess the potential association between antigen signal obtained on the automated platform (i.e., VITROS assay) and RT-PCR results (i.e., Ct values). The coefficients of variation [standard deviation/mean] × 100 (%)] of the quantitative signals provided by the VITROS automated AD test have been assessed by using three patient samples analyzed 10 times in a row. Data analysis was performed using GraphPad Prism^®^ software (version 9.0.0, California, CA, USA) and MedCalc^®^ software (version 14.8.1, Ostend, Belgium).

## 3. Results

### 3.1. Population

On the whole population, 96 patients (51.1%) had a positive RT-PCR (median Ct value: 22.3; min–max: 12.6–38.2). Among the symptomatic patients, 76 (64.4%) had positive RT-PCR results (median Ct value: 21.2; min–max: 12.6–38.2). Among the asymptomatic subjects, 20 (28.6%) had positive RT-PCR results (median Ct value: 25.9; min–max: 16.9–37.2). The Ct values were significantly lower in symptomatic patients compared to asymptomatic patients (*p* = 0.01, Appendix A). In symptomatic patients, the time from symptom onset to sampling did not differ between RT-PCR positive (median: 3 days, IQR: 2–4 days) or negative patients (median: 3 days, IQR: 2–5 days) (*p* = 0.31).

### 3.2. RAD Tests Sensitivity

The RAD tests were mostly effective to identify RT-PCR positive patients with higher viral loads (i.e., Ct values <25), with a sensitivity of 93.1% for the Biotical and the Panbio assays and of 96.6% for the Healgen and Roche assays at this threshold for positivity (Figure 1 and Appendix A). The Biotical, the Panbio, the Healgen, and the Roche RAD tests missed 4 (including one asymptomatic patient with a Ct value of 23.4), 4, 2, and 2 patients with Ct values ≤25, respectively (Figure 1 and Appendix A). For Ct values ranging from 25 to 30, RAD tests missed from 6 (31.6%, Healgen assay, reading at 20 min) to 13 (68.4%, Biotical assay) RT-PCR positive patients (Figure 1 and Table 2). Three asymptomatic subjects (i.e., Ct values of 25.5, 26.3, and 29.3) were not detected by any of the RAD tests. Discrepancies were observed between the different reading times. One Panbio result was positive after reading at 15 min (Ct value = 28.7) but turned negative at 20 min. On the other hand, one Panbio result was negative after reading at 15 min (Ct value = 26.4) but turned positive after 20 min. Some discordances were also noted with the Healgen assay. Five negative results at 15 min turned positive at 20 min. No discordance was observed with the Roche assay. For RT-PCR results with a Ct value >30, the number of RT-PCR positive patients detected was even lower (Table 2).

### 3.3. Ortho Sensitivity

The VITROS automated AG test showed a sensitivity of 100% for Ct values ≤33 (Table 2). Considering a Ct value of 35, the sensitivity slightly decreased to 96.4% (95%CI: 89.8–99.3%) (Appendix A). Additionally, an excellent correlation between the antigen signal obtained on the VITROS automated assay and Ct values was observed (R^2^ = 0.94; *p* <0.0001). The precision of the assay was excellent with within-run CVs of 2.3%, 1.8%, and 3.1% at Ct values of 20.7, 23.4, and 30.9, respectively.

### 3.4. Antigen Detection Tests Specificity

The specificities for the Biotical, the Panbio, the Healgen, and the Roche RAD were 98.9%, 100%, 97.8%, and 100% when results were read after 10–15 min (depending on the test) in samples rendered negative by the RT-PCR. Due to the discrepancies in some samples between the different reading times, it decreased to 96.8% for the Healgen at 20 min. When considering a Ct value of 33 as a surrogate of SARS-COV-2 contagiousness, specificities decreased for all RAD tests (Table 2). For the VITROS automated assay, none of the negative RT-PCR samples had a result above the cut-off signal of 1, meaning that the specificity reached 100%. Results were the same when considering a Ct value of 33 as the threshold for negativity.

### 3.5. Negative Predictive Value, Positive Predictive Value, and Accuracy

For the RAD tests, simulations showed that, at low disease prevalence, e.g., 2%, the PPV was low (from 24.6% to 64.5%) while the NPV was high (from 99.5% to 99.8%). At high disease prevalence, e.g., 25%, the PPV increased to 84.2 to 96.7% while the NPV decreased to 92.4 to 96.2%. The automated AD test, with its sensitivity and specificity of 100% at the cut-off Ct of 33, showed a PPV and NPV, and an accuracy of 100% (95% CI: 98.1% to 100%), respectively (Appendix A).

## 4. Discussion

This study compared and analyzed the clinical performance of 5 AD tests, including the automated assay from Ortho Clinical Diagnostics. The RAD tests were most effective to identify RT-PCR positive symptomatic patients or asymptomatic subjects with higher viral loads (i.e., Ct values ≤25). The sensitivity of these RAD tests for samples with a Ct values ≤25 was 93.1% for the Biotical and the Panbio assays, while it was 96.6% for the Healgen and the Roche assays. Only the Biotical assay missed an asymptomatic patient with a Ct value of 23.4. For Ct values comprised between 25 and 30, all RAD tests were not able to detect three asymptomatic subjects with Ct values of 25.5, 26.3, and 29.3, respectively. Sensitivities decreased proportionally with higher Ct values. The best sensitivity was observed with the Healgen assay (i.e., 88.8% for samples with Ct values ≤33). The VITROS automated assay outclassed all RAD tests with a sensitivity and a specificity of 100% for Ct values up to 33 (Table 2). The fact that the VITROS assay presented better performance compared to RAD tests is certainly related to an increased limit of quantification made possible by the chemiluminescent technology used.

In the literature, we identified at least four studies that evaluated the Panbio COVID-19 Ag rapid test [8,12,14,17,22]. In a population of 341 patients, Fenollar et al. found a specificity of 94.9% and a good sensitivity but only for samples with Ct values <25 (i.e., 94.2 to 100%) [12]. In a smaller cohort of 51 RT-PCR positive patients, Lanser et al. found a sensitivity of 85.8% in Ct values <25 [3]. Albert et al. confirmed a high specificity (i.e., 100%) with an overall sensitivity of 79.6% in a population of 54 RT-PCR positive patients. For Ct values <25, the sensitivity increased above 95% [8]. The same conclusion was found by Mak et al., using only 35 specimens [17]. The lower sensitivity observed with higher Ct values is in line with our evaluation (Table 2). Krüttgen et al. evaluated the Roche SARS-CoV-2 Rapid Antigen Test on a population of 75 patients with a positive RT-PCR and 75 with a negative RT-PCR and found a specificity of 96% and a 100% specificity in samples with Ct values lower than 25 [13]. Using the same assay, Salvagno et al. found a sensitivity ranging from 97 to 100% for specimens with Ct values <25, but lower sensitivities for higher Ct values (i.e., 50–81% for Ct values of 25 to <30) [23]. These results are also consistent with the data obtained in this study. Of note, this study is the first to evaluate the Biotical and Healgen RAD tests.

Our study is also the first to evaluate the new automated AD test provided by Ortho Clinical Diagnostics. Compared to RAD tests, the sensitivity of the test was 100% for Ct values up to 33, and the specificity of the test was 100% (Table 2). The assay only missed three patient samples for Ct values up to 35 (i.e., at Ct values of 33.2, 34.1, and 34.5) (Appendix A). Interestingly, we observed an excellent correlation between the antigen signal and Ct values obtained by RT-PCR (Figure 2). Hirotsu et al. evaluated another automated AG test, namely the LUMIPULSE SARS-CoV-2 Ag kit (Fujirebio, Tokyo, Japan), performed on a LUMIPULSE G600II chemiluminescent assay [29]. Out of 58 RT-PCR positive patients, 26 (44.8%) were negative on the LUMIPULSE. The specificity of the test ranged from 97.3% to 99.6% [29,30]. They also observed a weaker correlation with viral load (i.e., R^2^ = 0.77) compared to our evaluation with the VITROS automated assay (i.e., R^2^ = 0.94), which in our hands, showed excellent clinical performance.

In the scientific literature, there is an ongoing debate regarding the Ct value corresponding to the threshold of infectivity (i.e., patient considered as contagious) [26]. Bullard et al. showed that SARS-CoV-2 Vero cell infectivity was only observed for RT-PCR Ct value <24 in samples obtained less than eight days since symptom onset [4]. Albert et al. also found that SARS-CoV-2 could not be cultured from positive RT-PCR samples with Ct value >25. The same observation was made for samples positive using RT-PCR but negative using the Panbio RAD test (*n* = 11) [8]. However, in a larger cohort, Singanayagam et al. reported that a Ct value of 25 was still correlated to 80% of positive cultures and that only 8% of samples had a positive culture for Ct value >35 [28]. Jaafar et al. also showed that up to 70% of cultures were positive for a Ct value of 25, and only 3% of cultures were positive for a Ct value >35 [25]. La Scola et al. found that patients with Ct value >33–34 are not contagious because of the low number of positive cultures [27]. This is consistent with the Centers for Disease Control and Prevention (CDC) recommendations, which propose a Ct value of 33 as a surrogate of contagiousness [24]. Based on these studies and on CDC recommendations, we calculate the performance of the five AG tests at the different proposed cut-offs for contagiousness (Table 2, Appendix A).

The main advantages of RAD tests are their rapidity, and their ease of interpretation, and the limited technical skill/infrastructure requirements. Moreover, the use of RAD tests in mass screening programs could decrease the burden on laboratories that have been overwhelmed during the last COVID-19 pandemics [9]. Nevertheless, it is important to keep in mind that even with a high viral load (i.e., Ct values <25), a RAD test can result in false-negative estimates in both symptomatic patients and asymptomatic subjects [12,13,17,20,21,22,23]. Mass community screening would, therefore, require the use of more sensitive techniques.

The performance of the VITROS automated assay outclassed the ones of RAD tests. Based on the categorization proposed by Kost and considering a Ct value of 33 [31], the VITROS automated assay can be placed in the “Tier 3” category, meaning that the assay has high sensitivity and specificity (i.e., 100%), leading to high NPV and PPV whatever the prevalence of the disease (Appendix A).

In most laboratories using RT-PCR for diagnosis, the VITROS automated assay could be used as a routine high-throughput test in a clinical setting with an approximative cadence of 150 samples per hour. More importantly, according to the CDC cut-off, the VITROS automated assay may replace the RT-PCR for the detection of positive cases thanks to its excellent performance at the Ct value of 33. However, compared to RAD tests that can be used in any facility, an automated test obviously requires specialized equipment.

These analytical and clinical performance have to be interpreted together with the prevalence of the disease. While some studies have discussed the importance of the disease prevalence for serological testing [31], few have discussed the importance of the disease prevalence in the performance of AD tests. Simulations showed that, at low disease prevalence, e.g., 2%, the PPV was low (24.6% to 64.5%) while the NPV was high (99.5% to 99.8%). At high disease prevalence, e.g., 25%, the PPV increased from 84.2% to 96.7% while the NPV decreased from 92.4% to 96.2%. Consequently, RAD tests are not appropriate for mass community screening since they will lead to a high rate of false-positive and negative results. On the opposite, the VITROS assay showed remarkable performance with PPV and NPV of 100% both.

While this study confirms that RAD tests have limited performance and should only be used in facilities where access to RT-PCR or automated AD is not possible within an appropriate timeframe, further studies are needed to confirm our data on a larger population. In addition, a global consensus is also needed to define the Ct value that could be used as a surrogate of infectivity and contagiousness [2].

## 5. Conclusions

The RAD tests showed an acceptable sensitivity only for samples with Ct values corresponding to higher viral loads (i.e., <25). However, even with such high viral loads, some samples were miscategorized both from symptomatic patients and asymptomatic subjects. For Ct values between 25 and 30, only the VITROS automated assay showed a sufficient sensitivity. If considering a Ct value of 33 as a surrogate of contagiousness, the VITROS automated test performed the best (100% sensitivity) while RAD tests reported insufficient sensitivity. The highest specificity was also observed using the VITROS automated assay (i.e., 100%), which could be proposed as a first-line testing method for the detection of potential contagious cases.

## Figures and Tables

**Figure 1 jcm-10-00265-f001:**
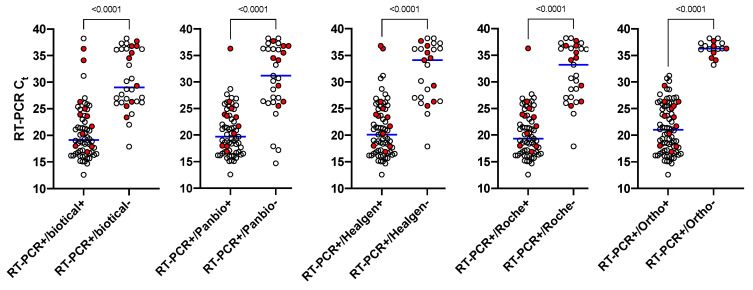
Graphical representation of positive and negative antigen results according to RT-PCR Ct values. A significant difference in the Ct value is observed between the positive and negative tests for each antigen method. Importantly, only the automated antigen shows no overlap between Ct values obtained for positive and negative samples. Samples from asymptomatic subjects are highlighted in red.

**Figure 2 jcm-10-00265-f002:**
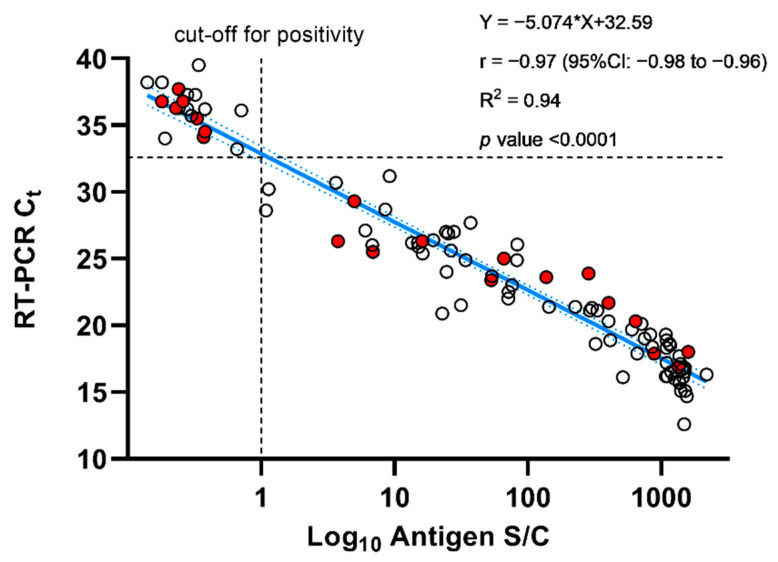
Linear regression of Ct values obtained by RT-PCR versus the amount of antigen (log_10_ transformed S/C results) obtained on the VITROS assay. Samples from asymptomatic subjects are highlighted in red.

**Table 1 jcm-10-00265-t001:** Characteristics of antigenic tests used in this study.

Test	Manufacturer	Process	Target	Technological	Procedure
Biotical SARS-CoV-2 Ag card	Biotical health	Manual	Nucleocapsid	Immunochromatographic	50 parts sample: 50 parts buffer; reading at 10 min. Positive if both control and test line are present. Negative if only the control band is present
Panbio™ COVID-19 Ag Rapid Test Device	Abbott	Manual	Nucleocapsid	Immunochromatographic	46 parts sample: 54 parts buffer; reading between 15 and 20 min. Positive if both control and test line are present. Negative if only the control band is present
Coronavirus Ag Rapid Test Cassette	Healgen Scientific	Manual	Nucleocapsid	Immunochromatographic	50 parts sample: 50 parts buffer; reading between 15 and 20 min. Positive if both control and test line are present. Negative if only the control band is present
Roche SARS-CoV-2 Rapid Antigen Test	Roche Diagnostics	Manual	Nucleocapsid	Immunochromatographic	50 parts sample: 50 parts buffer; reading between 15 and 30 min. Positive if both control and test line are present. Negative if only the control band is present
VITROS Immunodiagnostic Products SARS-CoV-2 Antigen test	Ortho Clinical Diagnostics	Automated	Nucleocapsid	Chemiluminescence	80 parts sample: 20 parts buffer. Positive if signal ≥1. Negative if signal <1. Time first result = 48 min, ≅150 samples per hour

**Table 2 jcm-10-00265-t002:** Sensitivity and specificity of RAD and automated antigen tests across different ranges of RT-PCR Ct values. Results in grey provide results of all RT-PCR values, including those with a Ct value >33 (min–max range: 12.6–38.2) for sensitivity and specificity.

Sensitivity		No. of Positive Patients							
Ct Range	*n*	Biotical (10 min)	Panbio (15 min)	Panbio (20 min)	Healgen (15 min)	Healgen (20 min)	Roche (15 min)	Roche (30 min)	Ortho
<15	2	2 (100%)	1 (50.0%)	1 (50.0%)	2 (100%)	2 (100%)	2 (100%)	2 (100%)	2 (100%)
>15–20	34	33 (97.1%)	32 (94.1%)	32 (94.1%)	33 (97.1%)	33 (97.1%)	33 (97.1%)	33 (97.1%)	34 (100%)
>20–25	22	19 (86.4%)	21 (95.5%)	21 (95.5%)	21 (95.5%)	21 (95.5%)	21 (95.5%)	21 (95.5%)	22 (100%)
>25–30	19	6 (31.6%)	10 (52.6%)	10 (52.6%)	11 (57.9%)	13 (68.4%)	10 (52.6%)	10 (52.6%)	19 (100%)
>30–33	3	1 (33.3%)	0 (0.0%)	0 (0.0%)	2 (66.7%)	2 (66.7%)	0 (0.0%)	0 (0.0%)	3 (100%)
Total <33 ^†^ [95% CI]	80	61 (76.2%) [65.4–85.1%]	64 (80.0%) [69.6–88.1%]	64 (80.0%) [69.6–88.1%]	69 (86.3%) [76.7–92.9%]	71 (88.8%) [79.7–94.7%]	66 (82.5%) [72.4–90.1%]	66 (82.5%) [72.4–90.1%]	80 (100%) [95.5–100%]
>33	16	3 (18.8%)	1 (6.3%)	1 (6.3%)	1 (6.3%)	3 (18.8%)	1 (6.3%)	1 (6.3%)	0 (0.0%)
Total (*n*, %) [95% CI]	96	64 (66.7%) [56.3–76.0%]	65 (67.7%) [57.4–76.9%]	65 (67.7%) [57.4–76.9%]	70 (72.9%) [62.9–81.5%]	74 (77.1%) [67.4–85.1%]	67 (69.8%) [59.6–78.8%]	67 (69.8%) [59.6–78.8%]	80 (83.3%) [74.4–90.2%]
**Specificity**		**No. of Negative Patients**							
Neg. RT-PCR ^‡^ [95% CI]	108	104 (96.3%) [90.8–99.0%]	107 (99.1%) [95.0–100%]	107 (99.1%) [95.0–100%]	105 (97.2%) [92.1–99.4%]	102 (94.4%) [88.3–97.9%]	107 (99.1%) [95.0–100%]	107 (99.1%) [95.0–100%]	108 (100%) [96.6–100%]
Neg. RT-PCR [95% CI]	92	91 (98.9%) [94.1–99.9%]	92 (100%) [96.1–100%]	92 (100%) [96.1–100%]	90 (97.8%) [92.4–99.7%]	89 (96.7%) [90.8–99.3%]	92 (100%) [96.1–100%]	92 (100%) [96.1–100%]	92 (100%) [96.1–100%]

^†^ The cut-off of 33 for the number of Ct has been chosen according to the Common Investigation Protocol for Investigating Suspected SARS-CoV-2 Reinfection of the Centers for Disease Control; ^‡^ Ct > 33 have been considered negative according to the Common Investigation Protocol for Investigating Suspected SARS-CoV-2 Reinfection of the Centers for Disease Control and Prevention.

## Data Availability

The data presented in this study are available on request from the corresponding author. The data are not publicly available due to ethical issue.

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
