# Peer review of "Head-to-Head Comparison of Rapid and Automated Antigen Detection Tests for the Diagnosis of SARS-CoV-2 Infection"

_jcm, 2021, doi:10.3390/jcm10020265_

Round 1

Reviewer 1 Report

Review of Favresse et al, “Head-to-head comparison of Rapid and Automated Antigen Detection Tests for the Diagnosis of SARS-CoV-2 Infection” for Journal of Clinical Medicine.

Overall comments:

The manuscript describes a comparative study where several commercially available tests for SARS-CoV-2 were examined, and compared to known RT-PCR results.  The authors use the data to calculate the sensitivity and specificity of the tests, identifying the automated test as the best performing.  While there is some value in this manuscript, the sample size is currently too low to draw any meaningful conclusions.  Additionally, the manuscript requires substantially rewriting to improve its legibility.  In its current form, the manuscript requires substantial revisions before being suitable for publication.

Specific comments:

Introduction – the introduction needs to be expanded to introduce the different tests used.  More citations of the current literature, particularly in regard to previous published comparative studies (which are cited within the conclusion) are required.

Materials and Methods – can the authors whether their use of the same viral transport media (line 81) might have impacted upon the results? What was the rational behind using only these 5 tests?  There are plenty more on the market which have better reported sensitivities and specificities than the ones used.  Additionally, all of these test use the Nucleocapsid only to determine positivity, while the authors use a PCR test that uses the envelope proteins.  The difference between these tests needs to be commented upon.  I am slightly confused as to why a third operator was required to interpret the results of the RAD tests, when they all rely upon the presence or absence of a band?  Surely it is simple enough to determine whether there is 1 band or 2?

Results – the authors should comment upon why differences in incubation time might alter the results.  Table 2, sup Table 1 and supp Table 2 are broadly speaking identical and provide minimal to no new information, can they not be combined into 1 table?  The authors have not commented on why the VITROS was the best performing.  It should also be mentioned in any observations that the need for specialised equipment for the Ortho may offset any improved performance.  

Discussion – the discussion is currently quite limited.  Most of it would be better placed in the introduction to give an overview to the topic.  The authors should actually comment on the tests they performed more and why the automated test performed so much better.  Line 177 references references 10, 12 and 16, yet the following text references 2, 12 and 16. Is the Ortho test that much faster than conventional RT-PCR? Most diagnostic laboratories using appropriate kits can process over 300 samples in an hour by RT-PCR. 

General writing – I cannot recommend any more strongly that the authors substantially rewrite the manuscript to improve its legibility or have a native English speaker rewrite it.  In its current form, it is unacceptable for publication.   

Author Response

Dear reviewer 1,

You will find here the changes made on the basis of your comments.

Review of Favresse et al, “Head-to-head comparison of Rapid and Automated Antigen Detection Tests for the Diagnosis of SARS-CoV-2 Infection” for Journal of Clinical Medicine.

Overall comments:

The manuscript describes a comparative study where several commercially available tests for SARS-CoV-2 were examined, and compared to known RT-PCR results. The authors use the data to calculate the sensitivity and specificity of the tests, identifying the automated test as the best performing. While there is some value in this manuscript, the sample size is currently too low to draw any meaningful conclusions. Additionally, the manuscript requires substantially rewriting to improve its legibility. In its current form, the manuscript requires substantial revisions before being suitable for publication.

Specific comments:

Introduction – the introduction needs to be expanded to introduce the different tests used.

  • We introduced the different tests used in the introduction.

More citations of the current literature, particularly in regard to previous published comparative studies (which are cited within the conclusion) are required.

  • We added more citations in the introduction.
  • However, we decided to stay brief in the introduction because we are convinced that the comparison of the performance of the assays described in this study with those of the literature should only be present in the discussion part. We also refer to JCM Guidelines for Authors that recommend a Brief introduction with the citation of key publications (https://www.mdpi.com/journal/jcm/instructions). We are thus in line with the editorial style.

Materials and Methods – can the authors whether their use of the same viral transport media (line 81) might have impacted upon the results?

  • All analyses were performed according to manufacturer’s recommendations and the same tube was used for both RT-PCR and antigenic assessments. Two sentences have been added this described this:

See page 2, line 75 and 92-93.

  • All analyses were performed under a chemical host. Technicians wore protective gloves, and used disposables tips directly placed in a chemical trash when the sample has been distributed.
  • The sample was distributed under two distinct chemical hosts (one for the RT-PCR and one for antigen detection tests), at two different moments.
  • The fact that the same VTM might have impacted our results is therefore very unlikely.

What was the rationale behind using only these 5 tests?

  • We agree with you that there are a lot of different antigen detection tests than could have been tested in our study. Currently, there is approximately 100 different RAD tests on the market (CE-IVD) (https://www.finddx.org/covid-19/pipeline/?avance=Commercialized&type=all&test_target=Antigen&status=CE-IVD&section=show-all&action=default). A sentence has been added in the introduction.
  • In our evaluation, we were not able to perform our analysis on more than 5 assays for mainly one reason: a limitation related to sample volume (because we wanted to compare all the assays by using the same patient’s sample). The fact that these devices are costly and that analyses are time-consuming represented another reason to only focus our report on 5 different tests. We also wanted to include antigen assays that are widely available on the market (i.e. Abbott and Roche assays). The two others are tests widely available regionally but are all CE-marked and distributed in the European market.
  • See page 2, lines 87-93.

There are plenty more on the market which have better reported sensitivities and specificities than the ones used.

  • A former Cochrane review found an average sensitivity of only 56.2% (95% CI 29.5 to 79.8%) for rapid antigen detection assays based on 8 different evaluations (Dinnes et al. 2020 10.1002/14651858.CD013705.). The reference has been added in the manuscript.
  • While performing our literature review, the sensitivity and specificities of the RAD tests used were in line with previous reports (see the discussion part; for Abbott and Roche given that data about biotical and Healgen were not found in other papers).
  • We also want to point out that sensitivities are mainly related to the viral load. In our evaluation, we compared antigen detection tests to each other, in relation to the viral load (i.e. Ct values). Sensitivities for RAD tests were high for Ct values <25 but were lower if considering higher Ct values. This may explain why some studies have reported higher sensitivity.

Additionally, all of these tests use the Nucleocapsid only to determine positivity, while the authors use a PCR test that uses the envelope proteins. The difference between these tests needs to be commented upon.

  • Most of the antigen detection tests on the market are using the nucleocapsid target (https://www.finddx.org/covid-19/pipeline/?avance=Commercialized&type=all&test_target=Antigen&status=CE-IVD&section=show-all&action=default). A sentence has been added (page 2, lines 88-89).

‘The target of the 5 AD tests is the nucleocapsid; the main antigen used in available AD tests on the market (https://www.finddx.org/covid-19/pipeline/?avance=Commercialized&type=all&test_target=Antigen&status=CE-IVD&section=show-all&action=default).”

  • We were only able to compare the results of antigen detection tests with the RT-PCR method currently available in our laboratory.
  • The target of the PCR is RNA while the target of the antigen detection tests are proteins.
  • Studies that compared the performance of antigen detection tests according to various RT-PCR technics are welcomed to attest this point. However, Puck B. van Kasteren has addressed the question of the performance of different RT-PCR kits on the markets and has concluded that both show similar performances. Thus, we do not expect (but cannot confirm based on our data) that different results will be obtained with other RT-PCR assays.

 I am slightly confused as to why a third operator was required to interpret the results of the RAD tests, when they all rely upon the presence or absence of a band?  Surely it is simple enough to determine whether there is 1 band or 2?

  • When the band intensity if high, no reading problem are reported. In such situation, we agree with this reviewer that it is simple enough to determine whether there is 1 band or 2.
  • However, the positivity of a band is sometimes very difficult to apprehend (close to the limit of detection of the test) and may therefore become reader-dependent. Some readers would read the test as positive (slight band appearance), while others would say that the test in negative.
  • Therefore, to be more objective, all the RAD tests were read by several readers.
  • Additional information has been provided. “Two independent operators (CG and MD) interpreted the results of the 4 RAD tests. In case of disagreement (i.e. especially for low band intensity), a third blinded operator (JD) interpreted the result to find a consensus result”

Results – the authors should comment upon why differences in incubation time might alter the results.

  • Incubation times are data provided by manufacturers. We only strictly applied manufacturer’s recommendations while performing the tests. We suppose that these different timing have been determined to optimize test performance. A sentence has been added. “All analyses were performed according to manufacturer’s recommendations.”
  • If the manufacturer claim that the reading should be performed at 10 min. We read at 10 min.
  • If the manufacturer claim that the reading should be performed between 15 and 30 min. We chose to read at both 15 and 30 min (for more objectivity).
  • The fact that some discrepancies might occur by reading at the different times recommended by the manufacturer is an important information for users and patients.

 Table 2, supp Table 1 and supp Table 2 are broadly speaking identical and provide minimal to no new information, can they not be combined into 1 table? 

  • The main message of our paper is to compare antigen detection tests at the CDC cut-off of 33.
  • Only the Table 2 would be present in the manuscript. The 2 other tables would be supplementary material (not directly visible in the manuscript).
  • Moreover, merging all tables together would result in a Table with too much information which will not stand in one page and would complicate the reading.
  • We therefore prefer to keep the actual configuration.

The authors have not commented on why the VITROS was the best performing. It should also be mentioned in any observations that the need for specialized equipment for the Ortho may offset any improved performance.

  • Indeed, we add a comment in the discussion to attest these 2 points. “The fact that the VITROS assay present better performance compared to RAD tests is certainly related to an increased limit of quantification made possible by the chemiluminescent technology used.” And “Compared to RAD tests that can be used in any facilities, an automated test obviously required a specialized equipment.”

Discussion – the discussion is currently quite limited.  Most of it would be better placed in the introduction to give an overview to the topic.

  • Some references are now cited already in the introduction.
  • A specific point about the disease prevalence was also added in the discussion. “These analytical and clinical performance have to be interpreted together with the prevalence of the disease. While some studies have discussed the importance of the disease prevalence for serological testing (31) few have discussed the importance of the disease prevalence in the performance of AD tests. Simulations showed that, at low disease prevalence, e.g. 2%, the PPV was low (24.6% to 64.5%) while the NPV was high (99.5% to 99.8%). At high disease prevalence, e.g. 25%, the PPV increased from 84.2% to 96.7% while the NPV decreased from 92.4% to 96.2%. Consequently, RAD tests are not appropriate for mass community screening since they will lead to a high rate of false positive and negative results. On the opposite, the VITROS assay showed remarkable performance with PPV and NPV of 100% both”.
  • The main novelty of our research was to present performance data about an automated assay. Currently, there is a very limited number of publications related to automated assays.

The authors should actually comment on the tests they performed more and why the automated test performed so much better.

  • Probably because the chemiluminescent technology used by the VITROS has a better sensitivity (i.e. limit of quantification) compared to immunochromatographic assays.
  • The sentence has been added.“The fact that the VITROS assay present better performance compared to RAD tests is certainly related to an increased limit of quantification made possible by the chemiluminescent technology used.”

Line 177 references references 10, 12 and 16, yet the following text references 2, 12 and 16. Is the Ortho test that much faster than conventional RT-PCR? Most diagnostic laboratories using appropriate kits can process over 300 samples in an hour by RT-PCR.

  • In Belgium, the number of laboratories than can performed 300 RT-PCR in one hour is extremely limited. Some laboratories do not even have access to the RT-PCR technology.
  • RT-PCR requires high workload, skillful operators, expensive instrumentation, and crucial biosafety measures.
  • In most situations, antigen detection (AD) tests are more rapid, less laborious, less expensive and require only a comparatively short training period. Also, antigenic testing will have lower impact on the budget of the social security.
  • We adapted our sentence in the introduction part and in the discussion part. “In most laboratories, the analysis throughput of RT-PCR is limited because it requires high workload, skillful operators, expensive instrumentation, and crucial biosafety measures” and “In most laboratories using the RT-PCR, the VITROS automated assay can really be used as a routine high-throughput test in the hospital setting with an approximative cadence of 150 samples per hour”.

General writing – I cannot recommend any more strongly that the authors substantially rewrite the manuscript to improve its legibility or have a native English speaker rewrite it.  In its current form, it is unacceptable for publication.

  • The manuscript has been reviewed by a native English speaker according to the recommendations of this reviewer.

Changes made by authors:

We have included the following Figure 2:

The Figure was just added after the already existing sentence ‘’Additionally, an excellent correlation between the antigen signal obtained on the VITROS automated assay and Ct values was observed (r = -0.97; P < 0.0001).”

Reviewer 2 Report

Please add 95% confidence intervals to the key metrics, e.g., Vitros sensitivity and specificity, in the Abstract. That will allow realistic assessment of uncertainty, despite the apparent perfect 100%'s stated for both.

Actually, it would be best to report confidence intervals for all metrics reported in the Abstract, if word count allows.

Please consider changing the terminology to "positive percent agreement" (for sensitivity) and "negative percent agreement" (for specificity).

The discussion of false negatives and false positives is scant. Kindly expand with consideration of prevalence, which is not mentioned in the paper.

Please see https://meridian.allenpress.com/aplm/article/doi/10.5858/arpa.2020-0443-SA/443500.

The reference deals with math analysis of COVID-19 tests, and is useful for organizing and relating them to clinical utility. It is Kost GJ,

Designing and Interpreting COVID-19 Diagnostics: Mathematics, Visual Logistics, and Low Prevalence

Archives of Pathology and Laboratory Medicine 2020;

https://doi.org/10.5858/arpa.2020-0443-SA   Note that a follow-up paper dealing with uncertainty and antigen tests will be posted online soon at the APLM website.

Next, categorize the evaluation results in terms of Tier 3, 2, and 1 in that paper. This will help the reader to put the results in perspective.

That is, antigen tests have striking weaknesses, yet the authors basically recommend the Vitros for "Ct up to 33," so what about below that, and what of the others? It would help to categorize the performance of the all of the tests and Ct's for clarity in terms of Tiers, which also relate to US and Canadian Emergency Use Authorization.

A few tune-ups, and this paper will be a very useful contribution regarding antigen tests. I reviewed the paper the same day I received it, so the word can get out promptly.

Once the tests are better categorized, maybe by Tiers 1, 2, and 3 as suggested, then the authors could wrap up the paper with some very useful suggestions regarding: a) for what prevalence they would be appropriate, and b) for what settings (e.g., urgent care, ER, hospital, limited-resource settings, airport departure screening, and so on).

Thank you for the opportunity of reviewing this paper.

Author Response

Dear reviewer 2,

Please find below a point-by-point response to your comments. We thank you for the time you have allocated to the review of this manuscript.

Comment 1: Please add 95% confidence intervals to the key metrics, e.g., Vitros sensitivity and specificity, in the Abstract. That will allow realistic assessment of uncertainty, despite the apparent perfect 100%'s stated for both. Actually, it would be best to report confidence intervals for all metrics reported in the Abstract, if word count allows.

  • We have added the 95% CI in the abstract as requested.

Comment 2: Please consider changing the terminology to "positive percent agreement" (for sensitivity) and "negative percent agreement" (for specificity).

  • Although we acknowledge that this comment makes sense and is appropriate, we think that, as the terms sensitivity and specificity are more frequently used, it is preferable to keep them for the sake of clarity.

Comment 3: The discussion of false negatives and false positives is scant. Kindly expand with consideration of prevalence, which is not mentioned in the paper. Please see https://meridian.allenpress.com/aplm/article/doi/10.5858/arpa.2020-0443-SA/443500. The reference deals with math analysis of COVID-19 tests, and is useful for organizing and relating them to clinical utility. It is Kost GJ, Designing and Interpreting COVID-19 Diagnostics: Mathematics, Visual Logistics, and Low Prevalence. Archives of Pathology and Laboratory Medicine 2020; https://doi.org/10.5858/arpa.2020-0443-SA 

Note that a follow-up paper dealing with uncertainty and antigen tests will be posted online soon at the APLM website. Next, categorize the evaluation results in terms of Tier 3, 2, and 1 in that paper. This will help the reader to put the results in perspective.

That is, antigen tests have striking weaknesses, yet the authors basically recommend the Vitros for "Ct up to 33," so what about below that, and what of the others? It would help to categorize the performance of the all of the tests and Ct's for clarity in terms of Tiers, which also relate to US and Canadian Emergency Use Authorization.

  • We thank the reviewer for this comment and for the reference. The prevalence of the disease is of upmost importance when considering the real-life performance of these tests. Therefore, we have performed simulation at different disease prevalence (i.e. 2, 5, 10, 15, 20 and 25%) to calculate the positive predictive value, the negative predictive value and the accuracy. The discussion has been adapted accordingly.

A few tune-ups, and this paper will be a very useful contribution regarding antigen tests. I reviewed the paper the same day I received it, so the word can get out promptly.

  • We thank you for reviewing this manuscript and for your encouraging comments.

Once the tests are better categorized, maybe by Tiers 1, 2, and 3 as suggested, then the authors could wrap up the paper with some very useful suggestions regarding: a) for what prevalence they would be appropriate, and b) for what settings (e.g., urgent care, ER, hospital, limited-resource settings, airport departure screening, and so on).

  • The VITROS automated assay can surely be placed in the “Tier 3” category if considering a Ct value of 33. The assay could therefore be used for mass community screening.
  • Based on the paper of Kost, the 4 RAD tests cannot even be categorized as “Tier 1” because the minimal sensitivity of 90% was not reached. Such rapid tests are therefore not recommended for mass community screening. They only can identify patients with high viral load (Ct <25) (sensitivity >90% but with a specificity <95 for the cut-off of Ct = 25). They can be used in limited-resource settings, but users need to be aware of the current limitations of these assays.
  • However, we think that the discussion about “Tiers” might be too confusing for the reader because it is quite a new concept and depend on each assay and on the Ct cut-off used. We rather prefer to benchmark assays based on their relative predictive values based on various disease prevalence (see supplemental table 3).

“These analytical and clinical performance have to be interpreted together with the prevalence of the disease. While some studies have discussed the importance of the disease prevalence for serological testing (31) few have discussed the importance of the disease prevalence in the performance of AD tests. Simulations showed that, at low disease prevalence, e.g. 2%, the PPV was low (24.6% to 64.5%) while the NPV was high (99.5% to 99.8%). At high disease prevalence, e.g. 25%, the PPV increased from 84.2% to 96.7% while the NPV decreased from 92.4% to 96.2%. Consequently, RAD tests are not appropriate for mass community screening since they will lead to a high rate of false positive and negative results. On the opposite, the VITROS assay showed remarkable performance with PPV and NPV of 100% both.”

Thank you for the opportunity of reviewing this paper.

Changes made by authors:

In the resubmitted article, we have also uploaded the Figure 2:

The Figure was just added after the already existing sentence ‘’Additionally, an excellent correlation between the antigen signal obtained on the VITROS automated assay and Ct values was observed (r = -0.97; P < 0.0001).”

Reviewer 3 Report

Thank you very much for this interesting manuscript!

Please explain, whether you collected one swap each for PCR and for AD analysis or one medium was used for both tests.

Please consider, whether it would not be better to speak of "Ct values" throughout the manuscript.

Please change in Acknowledgements "its" to "his".

Author Response

Dear reviewer 3,

We are pleased to respond below to each of your comments:

Comment 1: Thank you very much for this interesting manuscript!

  • We thank you for reviewing this manuscript and for your encouraging comments.

Comment 2: Please explain, whether you collected one swap each for PCR and for AD analysis or one medium was used for both tests.

  • We use the same tube for all analyses. This has been precised in the the M&M section.

“ The same tube was used for both RT-PCR and antigenic assessments. »

Comment 3: Please consider, whether it would not be better to speak of "Ct values" throughout the manuscript.

  • According to your suggestion Ct value(s) has been used throughout the manuscript.

Comment 4: Please change in Acknowledgements "its" to "his".

  • This has been corrected accordingly.

Changes made by authors:

In the resubmitted article, we have also uploaded the Figure 2:

The Figure was just added after the already existing sentence ‘’Additionally, an excellent correlation between the antigen signal obtained on the VITROS automated assay and Ct values was observed (r = -0.97; P < 0.0001).”

Round 2

Reviewer 1 Report

2nd review

Comments to authors

The authors have sufficiently altered the manuscript to take into account my previous comments.  I still believe the sample size is too small to warrant any conclusions to be drawn from the data, however if the authors are unable to expand this, then I have no further comments.  I commend the authors for substantially improving the legibility of the manuscript.  The following small language comments I have are below.

Line 27 –  “us” missing after “allowed”

44 – “do” not “does”

45 – “about” missing before “whether”

Line 56-57 – “remains” instead of “is still very”

58 – “to compare”

61 – delete “according”

197 – remove comma after “that”

257 – “and” after comma, replace the next “and” with “/”

269 – remove “the”, add “for diagnosis” after RT-PCR, replace “can really” with could

270 – replace “the hospital” with “a clinical”

Paragraphs 257 to 263 and 275 to 283 should be combined as they have the same conclusion about the inappropriateness of using RAD for population studies.

Author Response

Dear reviewer 1,

You will find here the changes made on the basis of your comments.

Review of Favresse et al, “Head-to-head comparison of Rapid and Automated Antigen Detection Tests for the Diagnosis of SARS-CoV-2 Infection” for Journal of Clinical Medicine.

Comments to authors:

The authors have sufficiently altered the manuscript to take into account my previous comments.  I still believe the sample size is too small to warrant any conclusions to be drawn from the data, however if the authors are unable to expand this, then I have no further comments.  I commend the authors for substantially improving the legibility of the manuscript. 

  • Thank you for your comment

The following small language comments I have are below.

Line 27 –  “us” missing after “allowed”

44 – “do” not “does”

45 – “about” missing before “whether”

Line 56-57 – “remains” instead of “is still very”

58 – “to compare”

61 – delete “according”

197 – remove comma after “that”

257 – “and” after comma, replace the next “and” with “/”

269 – remove “the”, add “for diagnosis” after RT-PCR, replace “can really” with could

270 – replace “the hospital” with “a clinical”

All these changes have been considered in the revised manuscript
